# GETTING A-ROUND GUARANTEES: FLOATING-POINT ATTACKS ON CERTIFIED ROBUSTNESS

## ABSTRACT

Adversarial examples pose a security risk as they can alter decisions of a machine learning classifier through slight input perturbations. Certified robustness has been proposed as a mitigation where given an input $\mathbf{x}$, a classifier returns a prediction and a certified radius $R$ with a provable guarantee that any perturbation to $\mathbf{x}$ with $R$-bounded norm will not alter the classifier's prediction. In this work, we show that these guarantees can be invalidated due to limitations of floating-point representation that cause rounding errors. We design a rounding search method that can efficiently exploit this vulnerability to find adversarial examples against state-of-the-art certifications in two threat models, that differ in how the norm of the perturbation is computed. We show that the attack can be carried out against linear classifiers that have exact certifiable guarantees and against neural networks that have conservative certifications. In the weak threat model, our experiments demonstrate attack success rates over 50% on random linear classifiers, up to 23% on the MNIST dataset for linear SVM, and up to 15% for a neural network. In the strong threat model, the success rates are lower but positive. The floating-point errors exploited by our attacks can range from small to large (*e.g.*, $10^{-13}$ to $10^3$) — showing that even negligible errors can be systematically exploited to invalidate guarantees provided by certified robustness. Finally, we propose a formal mitigation approach based on bounded interval arithmetic, encouraging future implementations of robustness certificates to account for limitations of modern computing architecture to provide sound certifiable guarantees.

## 1 INTRODUCTION

Robustness of modern image classifiers has come under scrutiny due to a plethora of results demonstrating adversarial examples—small perturbations to benign inputs that cause models to mispredict, even when such perturbations are not evident to the human eye Madry et al. (2018); Carlini & Wagner (2017); Szegedy et al. (2014); Goodfellow et al. (2015). If a learned model is used in critical applications such as self-driving cars, clinical settings or malware detection, such easily added perturbations can have severe consequences. As a result, research focus has shifted to training models robust to adversarial perturbations, that come endowed with *certified robustness*.

Mechanisms for providing robustness certification aim to bound a model $f$'s sensitivity to a certain level of perturbation. At a high level, such mechanisms return a radius $R$ around a test input $\mathbf{x}$ with a guarantee that for any $\mathbf{x}'$ within $R$ distance from $\mathbf{x}$, $f(\mathbf{x}) = f(\mathbf{x}')$. How $R$ is computed, whether it is sound and/or complete depends on the mechanism. For example, bound propagation Zhang et al. (2018); Wang et al. (2021) transfers the upper and lower bounds from the output layer to the input layer of a neural network, and gives a lower bound on the perturbation needed to flip the classification.

Given the extensive research on certified robustness, can such mechanisms protect against adversarial examples in practice? In this paper, we show that the limits posed by floating-point arithmetic invalidate guarantees of several prominent mechanisms and their implementations. Despite proofs of robustness guarantees, they all assume real numbers can be represented exactly. Unfortunately, this critical (implicit) assumption cannot hold on computers with finite number representations. Since floating-point (FP) numbers can represent only a subset of real values, rounding is likely to occur when computing robust guarantees and can cause overestimation of the certified radius $R$. Thus, adversarial examples may exist within the computed radius despite claims of certification.

We devise a rounding search method that can efficiently discover such adversarial examples in two threat models, that differ in how the norm of the perturbation is computed. Our method is inspired by the traditional adversarial example search methods such as PGD Madry et al. (2018) and C&W Carlini & Wagner (2017). However, we find that such existing methods do not effectively exploit the rounding of a certified radius as the search space they explore is large (*i.e.*, the number of examples to check becomes intractable due to the large number of floating-point values) and instances of inappropriate rounding do not necessarily follow model gradients. To this end, our method is different from these search methods in two aspects: (1) instead of relying on back propagation, it leverages the piecewise linear property of ReLU networks to find coarse-level perturbation directions; (2) it then searches in a much finer scale by sampling floating-point neighbors of a potential adversarial example. The first aspect allows us to narrow down the search space closer to the certified radius and efficiently find adversarial examples. The second aspect enables our search method to find adversarial examples with perturbation norms that are just smaller than the certified radius (*e.g.*, in the 13th decimal place), which PGD and C&W cannot find. Compared to other works that find robustness violations Jia & Rinard (2021); Zombori et al. (2020), our attack method is arguably stronger as it works on unmodified target models with unaltered instances as opposed to specially crafted models or instances. We discuss the potential impact of our attacks on robustness guarantees in Appendix F.

One's first intuition to mitigate the overestimation of certified radii exploited by the above attacks might be to adopt slightly more conservative radii (*e.g.*, using $R - \gamma$ for some positive constant $\gamma \ll 1$). Unfortunately, such radii are not in general sound and choosing $\gamma$ is inherently error prone. That is, we show that the amount of overestimation can depend on the data (*e.g.*, number of features) and model (*e.g.*, number of operations) and that attacking $R - 0.1$ is still possible. To this end, we propose a defense based on rounded interval arithmetic that has theoretical guarantees and can be easily integrated into mechanisms for computing certified radii. In summary our contributions are:

- We explore a class of attacks that invalidate the implementations of certified robustness (*i.e.*, find adversarial examples within the certified radius). Our attacks exploit rounding errors due to limited floating-point representation of real numbers.

- We devise a rounding search method that systematically discovers such adversarial examples under two threat models. The weak model assumes that attacks need only have floating-point norms that violate certifications (e.g., in the case where the norm is computed using common software libraries). The strong model makes no such assumption: the true (real-valued) norm of attacks must violate certifications (e.g., in the case where the library that computes the square root for the norm can represent a real value or its range).

- We show that our attacks work against exact certifications of linear models Cohen et al. (2019), and against a conservative certified radius returned by a prominent neural network verifier Wang et al. (2021) on a network. Our attack success rate differs between learners and threat models. In the weak threat model, our success rates are over 50% for random linear classifiers and 15% on an MNIST neural network model. In the strong threat model, the attack success rates are lower but are still non-zero. For all cases, in theory, the certification should guarantee a 0% success rate for such attacks within certified radii.

- We propose a defense based on rounded interval arithmetic, with strong theoretical and empirical support for mitigating rounding search attacks.

## 2 BACKGROUND AND PRELIMINARIES

Let input instance $\mathbf{x} = (x_1, x_2, \ldots, x_D)$ be a vector in $\mathbb{R}^D$ with $x_i$ denoting the $i$th component of $\mathbf{x}$. We consider classifiers $f$ mapping an instance in $\mathbb{R}^D$ to a binary class label in $\{-1, 1\}$ or to a $K$-class label in $[K] = \{1, \ldots, K\}$.

**Adversarial examples.** Given an input instance $\mathbf{x}$, a classifier $f$, and a target label $t \neq f(\mathbf{x})$, $\mathbf{x}'$ is a *targeted* adversarial example Szegedy et al. (2014) if $f(\mathbf{x}') = t$ where $\mathbf{x}'$ is reachable from $\mathbf{x}$ according to some chosen threat model. In the vision domain, it is common to assume that small $\ell_p$ perturbations to $\mathbf{x}$ will go unnoticed by human observers. In this paper we consider $\ell_2$ distance, *i.e.*, $\|\mathbf{x} - \mathbf{x}'\| \leq \Delta$ for some small perturbation limit $\Delta$. An adversarial example in the multi-class setting is *untargeted* if $t$ is not specified.

**Floating-point representation.** Floating-point values represent reals using three binary numbers: a sign bit b, an exponent e, and a significand $d_1 d_2 \ldots d_d$. For example, 64-bit (double precision) floating-point numbers allocate 1 bit for b, 11 bits for e, and 52 bits for the significand. Such a floating-point number is defined to be $(-1)^b \times (1.d_1 d_2 \ldots d_d)_2 \times 2^{e-1023}$. Floating points can represent only a finite number of real values. Hence, computations involving floating-point numbers often need to be rounded up or down to their nearest floating-point representation IEEE.

## 2.1 CERTIFIED ROBUSTNESS

A robustness certification for a classifier at input $\mathbf{x}$ is a neighborhood (typically an $\ell_2$ ball) of $\mathbf{x}$ on which classifier predictions are constant. Certifications aim to guarantee that *no perturbed adversarial examples exist in this neighborhood*, including "slightly" perturbed instances.

**Definition 1** *A pointwise robustness certification for a $K$-class classifier $f$ at input $\mathbf{x} \in \mathbb{R}^D$ is a real radius $R > 0$ that is sound and (optionally) complete:*

*(i)* [sound] $\forall \mathbf{x}' \in \mathbb{R}^D, \|\mathbf{x}' - \mathbf{x}\| \leq R \Rightarrow f(\mathbf{x}') = f(\mathbf{x})$.

*(ii)* [complete] $\forall R' > R, \exists \mathbf{x}' \in \mathbb{R}^D, \|\mathbf{x}' - \mathbf{x}\| \leq R' \wedge f(\mathbf{x}') \neq f(\mathbf{x})$.

For a given certification mechanism, we will distinguish the idealized certification radius $R$ (*i.e.*, the mapping of Definition 1 under the soundness condition) from a candidate radius $\tilde{R}$ that an implementation of this mechanism computes. As we will see, the latter may not be necessarily sound (or complete). We categorize certification mechanisms into three kinds depending on their claims.

**Exact certification mechanisms.** These mechanisms output sound and complete radii under ideal realization of $\mathbb{R}$ arithmetic. Binary linear classifiers $f(\mathbf{x}) = sign(\mathbf{w}^T \mathbf{x} + b)$ admit a certified radius $R = |\mathbf{w}^T \mathbf{x} + b| / \|\mathbf{w}\|$. Cohen *et al.* derive this radius and prove its soundness (Cohen et al., 2019, Proposition 4) and completeness (Cohen et al., 2019, Proposition 5) for real arithmetic.

**Conservative certification mechanisms.** These are mechanisms that output radii that are sound and not necessarily complete under real-valued arithmetic. Bound propagation aims to provide a certified lower bound of minimum distortion Zhang et al. (2018); Wang et al. (2021); Wong & Kolter (2018); Wang et al. (2018) to an input that would cause a label change.

**Approximate mechanisms.** Approximate certifications output random radii that under $\mathbb{R}$ are sound (or abstain), with high probability $1 - \alpha$, and that are not necessarily complete. Randomized smoothing Cohen et al. (2019) is an example of this approach.

## 3 ROUNDING SEARCH ATTACK

We now present a rounding search method that exploits floating-point rounding errors to find adversarial examples within a computed certified radius $\tilde{R}$.

**Threat model.** Like prior works on adversarial examples Carlini & Wagner (2017); Madry et al. (2018); Jia & Rinard (2021), we assume that the adversary has white-box access to a classifier $f$, and has white-box access to the certification mechanism that it can query with inputs $f$ and instance $\mathbf{x} \in \mathbb{R}^D$, and obtain a certified radius $\tilde{R}$ as an output.

Since there are floating-point rounding errors in the operations for computing a certification, the computed radius $\tilde{R}$ at an instance $\mathbf{x}$ could overestimate an intended sound (and possibly complete) radius $R < \tilde{R}$. This creates leeway for an adversary to find adversarial perturbations whose norms are less than or equal to the computed certified radius, but which can change the classifications of the model, *invalidating soundness of the computed certification*. Our work aims to find a systematic and efficient way to exploit these rounding errors.

A perturbation's norm $\|\boldsymbol{\delta}\| = \|\mathbf{x}' - \mathbf{x}\|$ must be estimated when evaluating the perturbation's success. This norm computation can also suffer floating-point rounding errors, and could be underestimated. To handle this possibility, we conduct attacks in two threat models, one weak one strong.

The weak model makes minimal assumptions on how certification is violated: an attack is ruled successful if the floating-point computation of $\|\boldsymbol{\delta}\|$ is smaller than or equal to $\tilde{R}$ (*i.e.*, $\|\boldsymbol{\delta}\| \leq \tilde{R}$). We note that this model represents settings where the norm is computed using common software libraries for computing operations on floating numbers (e.g., Numpy's 32-bit or 64-bit floating-point arithmetic).

The strong model does not make these assumptions, the true (real-valued) norm of attacks must violate certifications. This model considers a setting where the norm is computed using software packages that instead of returning a result that is potentially rounded can return a representation of a real-valued norm or its range. Since we cannot do real arithmetic on machines, we use the upper bound of the norm $\overline{\|\boldsymbol{\delta}\|}$ instead, which is computed with bounded interval arithmetic and is guaranteed to be greater than or equal to the true norm. That is, a successful attack satisfies $\overline{\|\boldsymbol{\delta}\|} \leq \tilde{R}$.

**Attack overview.** Consider a classifier $f$, input $\mathbf{x}$ and the corresponding computed radius $\tilde{R}$. A naïve way to search for an adversarial example would be to try all $\mathbf{x}'$ such that $\|\mathbf{x}-\mathbf{x}'\| \leq \tilde{R}$, checking whether $f(\mathbf{x}) \neq f(\mathbf{x}')$. Unfortunately this exhaustive search is computationally intractable (*e.g.*, there are $\approx 2^{17}$ floating points in a small interval such as $[10, 10 + 2^{-32}]$). We can avoid some futile search. For example, observe that instances in the gray area, as depicted in Figure 1, are unlikely to flip predictions, as they are in the opposite direction of the decision boundary. A key idea is to find a perturbation direction $\boldsymbol{\nu}$ that reaches the decision boundary in the shortest distance, and add a perturbation $\boldsymbol{\delta}$ in that direction to $\mathbf{x}$, to maximize our chance to flip the classifier's prediction with perturbation norm $\|\boldsymbol{\delta}\|$ (or $\overline{\|\boldsymbol{\delta}\|}$) less than or equal to $\tilde{R}$. This baseline method has several challenges. First, computation of perturbation direction $\boldsymbol{\nu}$ is not easy for NNs which do not typically have linear decision boundaries. To this end, for ReLU networks, we find a local linear approximation prior to computing the gradient for $\boldsymbol{\nu}$. Second,

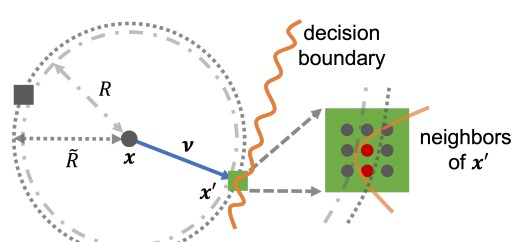

Figure 1: The search direction $\boldsymbol{\nu}$ (blue line) and search area (green area) for finding adversarial examples against a model, whose decision boundary is the orange line. $\mathbf{x}$ is the original instance, $\tilde{R}$ and $R$ are the computed and real-valued certified radii of the model on $\mathbf{x}$, $\boldsymbol{\delta} = \tilde{R}\boldsymbol{\nu}/\|\boldsymbol{\nu}\|$ is the adversarial perturbation in the search direction $\boldsymbol{\nu}$, instance $\mathbf{x}' = \mathbf{x} + \boldsymbol{\delta}$ is the seed for the green search area. Our rounding search method will sample $N$ floating-point neighbors $\boldsymbol{\delta}'$ of $\boldsymbol{\delta}$, and evaluate each $\mathbf{x} + \boldsymbol{\delta}'$ to check if any one of them can flip the classification of the model with $\|\boldsymbol{\delta}'\| \leq \tilde{R}$ or $\overline{\|\boldsymbol{\delta}'\|} \leq \tilde{R}$ (the red points in the green search area).

while $\boldsymbol{\nu}$ guides a search towards the decision boundary, the search may still be unable to exploit the leeway between the real certified radius $R$ and the computed certified radius $\tilde{R}$ to find certification violations. We address this challenge with a tightly-confined randomized floating-point neighborhood search. In summary, our attack proceeds as follows (depicted in Figure 1).

1. Find an adversarial perturbation direction $\boldsymbol{\nu}$ that reaches the decision boundary of classifier $f$ in the shortest distance, as a form of PGD attack Madry et al. (2018) (Section 3.1).

2. Compute perturbation $\boldsymbol{\delta}$ in the direction $\boldsymbol{\nu}$ within the computed certified radius $\tilde{R}$:

$$\boldsymbol{\delta} = \tilde{R}\boldsymbol{\nu}/\|\boldsymbol{\nu}\| \ . \tag{1}$$

3. Search for multiple floating-point neighbors $\boldsymbol{\delta}'$ of $\boldsymbol{\delta}$ with $\|\boldsymbol{\delta}'\| \leq \tilde{R}$ (or $\overline{\|\boldsymbol{\delta}'\|} \leq \tilde{R}$), and evaluate if any $\mathbf{x} + \boldsymbol{\delta}'$ can flip the classifier's prediction (Section 3.2).

### 3.1 ADVERSARIAL PERTURBATION DIRECTION

For linear models, direction $\boldsymbol{\nu}$ is a normal to the decision boundary's hyperplane $\mathbf{w}^T\mathbf{x} + b = 0$ and equals $\mathbf{w}$. The perturbation direction for neural networks is not as obvious as it is for linear models, as the decision boundary can be highly non-linear. In the rest of this section we describe our approach for finding $\boldsymbol{\nu}$ for neural networks with ReLU activations that we show to be effective in our experiments. A neural network with ReLUs can be represented as

$$F(\mathbf{x}) = (F_n \circ F_{n-1} \circ \cdots \circ F_1)(\mathbf{x})$$

where $F_i(\mathbf{x}) = \mathsf{ReLU}(\boldsymbol{\theta}_i^T \mathbf{x} + \hat{\boldsymbol{\theta}}_i)$. Here $\mathbf{x}$ and $\hat{\boldsymbol{\theta}}_i$ are vectors, $\boldsymbol{\theta}_i$ is a matrix, and the rectified linear (ReLU) activation function acts pointwise on a vector, returning a vector.

We use the fact that such networks are piecewise linear: therefore a (local) linear approximation at instance $\mathbf{x}$ is in fact *exact*. Then, one can find an adversarial example for $\mathbf{x}$ against this linear model as described above and use it to attack the original ReLU network.

**Warmup.** As a warmup, let us consider a network where ReLUs are all activated. For each node $\mathsf{ReLU}(z) = \max\{0, z\} = z$, and so the network is a combination of $K$ linear models where $K$ is the number of classes. That is,
$$F(\mathbf{x}) = \boldsymbol{\theta}^T \mathbf{x} + \hat{\boldsymbol{\theta}} \ ,$$
where $\boldsymbol{\theta}^T = \boldsymbol{\theta}_n^T \boldsymbol{\theta}_{n-1}^T \cdots \boldsymbol{\theta}_1^T$, and $\hat{\boldsymbol{\theta}} = \sum_{i=1}^n \left( \prod_{j=i+1}^n \boldsymbol{\theta}_j^T \right) \hat{\boldsymbol{\theta}}_i$. Note that $\boldsymbol{\theta}^T$ is a $K \times D$ matrix and $\hat{\boldsymbol{\theta}}$ is a column vector of length $K$. Each class $k$ corresponds to the linear model
$$F^k(\mathbf{x}) = \mathbf{w}_k^T \mathbf{x} + b_k \ ,$$
where $\mathbf{w}_k^T = (\boldsymbol{\theta}^T)_{k,\cdot}$ is the $k$th row of $\boldsymbol{\theta}^T$ and $b_k = \hat{\theta}_k$.

In order to change this model's classification from the original class $l$ to the target class $t \neq l$, we observe that one can attack the following model:
$$L(\mathbf{x}) = F^t(\mathbf{x}) - F^l(\mathbf{x}) = (\mathbf{w}_t^T - \mathbf{w}_l^T)\mathbf{x} + b_t - b_l \ .$$
This is a linear model, and $L(\mathbf{x}) < 0$ when $F(\mathbf{x})$ classifies $\mathbf{x}$ as $l$, $L(\mathbf{x}) > 0$ when $F(\mathbf{x})$ classifies $\mathbf{x}$ as $t$, so $L(\mathbf{x})$ has the decision boundary hyperplane $L(\mathbf{x}) = 0$. Hence, the most effective perturbation direction to change classification of $F(\mathbf{x})$ from $l$ to $t$, as before for linear models, is $\boldsymbol{\nu} = \mathbf{w}_t^T - \mathbf{w}_l^T$, which is the gradient of $L(\mathbf{x})$ with respect to $\mathbf{x}$.

**Linear approximation of ReLU networks.** ReLUs will all be activated when the weights and biases of each hidden layer of the network are positive, and all values of the input are also positive (*e.g.*, an image, whose pixel value is usually in the range $[0, 1]$). However, in practice this usually is not the case and some ReLUs will not be activated. For inactive ReLUs, we modify outgoing weights to zero in the calculation of the perturbation direction $\boldsymbol{\nu}$.

The overall process, LinApproxPerturbDir, is described in Algorithm 1 of Appendix A. It proceeds by first finding an exact (local) linear approximation $F'(\mathbf{x}) = \boldsymbol{\tau}^T \mathbf{x} + \hat{\boldsymbol{\tau}}$ where $\hat{\boldsymbol{\tau}} \leftarrow \sum_{i=1}^n \left( \prod_{j=i+1}^n \boldsymbol{\tau}_j^T \right) \hat{\boldsymbol{\theta}}_i$ using the notation in the pseudo-code. The weights of $F'$ are equal to weights of $F$ for internal nodes where $F(\mathbf{x})$ activated the corresponding ReLUs, otherwise they are set to 0. Specifically, we zero out columns of matrix $\boldsymbol{\theta}_i^T$ when the corresponding elements of mask $\mathbf{m}_i$ are zero. Given these weights, LinApproxPerturbDir computes $\boldsymbol{\nu}$ as explained in the warmup. This direction corresponds to a gradient of the network's target minus current class scores, with respect to the instance $\mathbf{x}$.

**Projected gradient descent for ReLU networks.** Given $\boldsymbol{\nu}$ as output by Algorithm 1 of Appendix A and a computed certified radius, one could compute adversarial perturbation $\boldsymbol{\delta}$ in direction $\boldsymbol{\nu}$ close to the certified radius as in Equation 1. However, the resulting $\mathbf{x}' = \mathbf{x} + \boldsymbol{\delta}$ may activate different ReLUs of $F$ than $\mathbf{x}$. Hence, the linear approximation $F'$ on $\mathbf{x}'$ may be different to $F'$ on $\mathbf{x}$: these approximations are only exact in local neighborhoods. To this end we perform a search by iteratively updating $\mathbf{x}'$ and invoking LinApproxPerturbDir until an adversarial example within the input domain $[V_{\min}, V_{\max}]$ is found or the procedure times out. Algorithm 2 of Appendix A describes this procedure, which we refer to as ReluPGD. The algorithm iteratively performs the following: computes the gradient of the network's linearization at the current iteration, rescales to the step size $s$, clips the perturbation to the domain constraint, applies the perturbation.

**Remark 1** *Note that $\tilde{R}$ may not be given, as is the case for some network verifiers that instead of returning $\tilde{R}$, take $F$, $\mathbf{x}$ and some $R$ as input and either certify $R$ or not. In this case, we need to search for the smallest perturbation in the direction of $\boldsymbol{\nu}$ to find such an $R$ to attack. Hence, in Algorithm 2 of Appendix A we use $s$ as an input, which is set to a small initial value (e.g., $10^{-5}$ in our experiments) so that $\boldsymbol{\nu}$ can be updated frequently. If a $\tilde{R}$ is given, we can set it as a threshold value to stop the algorithm, that is, the algorithm should stop when the total perturbation norm reaches $\tilde{R}$.*

## 3.2 ROUNDING SEARCH

Given the direction $\boldsymbol{\nu}$ and the computed certified radius $\tilde{R}$, an adversarial perturbation $\boldsymbol{\delta}$ can be computed using Equation 1, and $\mathbf{x}' = \mathbf{x} + \boldsymbol{\delta}$ should give an adversarial example so that $F(\mathbf{x}) \neq F(\mathbf{x}')$.

If the accumulated rounding errors are large, $\boldsymbol{\delta}$ can be sufficient to conduct a successful attack (*e.g.*, for neural networks with many neurons). For some attacks, the rounding errors we exploit are much smaller, such as linear models with fewer operations. Hence, we create $N$ floating-point neighbors of $\boldsymbol{\delta}$ to explore more possibilities of robustness violations close to the decision boundary due to rounding errors. At a high level, each neighbor $\boldsymbol{\delta}'$ is constructed by using $\boldsymbol{\delta}$ as a seed and then, for each dimension, replacing the original value with a neighboring floating point that is either larger or smaller than it. For example, a neighbor of $[1.0, 1.0]$ can be $[0.9999999999999999, 1.0000000000000002]$. We provide the pseudo-code of the neighbors sampling procedure in Algorithm 3 of Appendix A. We call this algorithm Neighbor. The result is a set of $N$ neighboring perturbations. Then for each neighbor $\boldsymbol{\delta}'$ we test if $\mathbf{x} + \boldsymbol{\delta}'$ leads to an adversarial example (*i.e.*, flips the classifier's prediction) that is certified (*i.e.*, $\|\boldsymbol{\delta}'\| \leq \tilde{R}$ in the weak threat model, or $\overline{\|\boldsymbol{\delta}'\|} \leq \tilde{R}$ in the strong threat model).

## 4 ATTACK EXPERIMENTS

In this section we evaluate whether our rounding search attacks can find adversarial examples within a certified radius. We first consider linear classifiers and then neural networks. We evaluate certified radii obtained using the *exact* method for linear classifiers, and *conservative* Wang et al. (2021); Gurobi Optimization, LLC (2022) and *approximate* Cohen et al. (2019) certification mechanisms for neural networks. Since the computation of exact certification for linear classifier is $R = |\mathbf{w}^T\mathbf{x} + b|/\|\mathbf{w}\|$, we compute it ourselves using either 32-bit or 64-bit floating-point arithmetic in Numpy.

For linear classifiers, we will conduct attacks in both the weak and strong threat models. For neural networks, we will conduct attacks in the weak threat model. We show that our rounding search finds adversarial examples within certified radii for all of them. Our linear models are run on an Intel Xeon Platinum 8180M CPU, while our neural network models are run on a Tesla V100 16GB GPU.

**Baseline attack rates.** The baseline success rate for finding an adversarial example against a linear model within the radius defined in Section 2.1 should be 0% in both threat models, since the mechanism is exact: it claims to be both sound and complete. The baseline success rate for radii returned by conservative mechanisms should also be 0% since they too are claimed to be sound. Though randomized smoothing comes with a failure probability $\alpha \ll 1$ to account for sampling error in approximating a smoothed classifier, it does not explicitly take into account errors due to rounding.

**Model training.** We train (primal) linear SVM with sequential minimal optimization, by using corresponding modules of scikit-learn. Our linear classifiers are trained with $\ell_1$ regularization so that model weights are sparse, and perturbations are less likely to move images outside their legal domain (recall that the perturbation direction for a linear classifier is its weights $\boldsymbol{\nu} = \mathbf{w}$). All ReLU networks in this section are trained with the SGD optimizer using PyTorch, with momentum 0.9, learning rate 0.01, batch size 64, for 15 epochs. For some controlled experiments we require the weights and biases of the hidden layers to be positive (to activate all ReLUs). In this case weights and biases are clamped with the lower bound 0 after each step of training.

### 4.1 RANDOM LINEAR CLASSIFIERS

To evaluate the performance of our attack in an ideal scenario, we first conduct our attack on randomly initialized (binary) linear classifiers with randomly generated target instances: $f(\mathbf{x}) = sign(\mathbf{w}^T\mathbf{x} + b)$, where weights $w_i$ and bias $b$ are random values drawn from the range $[-1, 1]$, $\forall i \in [D] = \{1, \ldots, 100\}$. Each value is represented with either 32-bit or 64-bit floating-point precision. For each dimension, we test 10 000 randomly initialized models. For each model we choose one instance $\mathbf{x}$ to attack, where each component $x_i$ is drawn randomly from $[-1, 1]$. Hence, attack success rate measures the number of models out of 10 000 for which a random instance can result in a successful attack. For each combination of $(\mathbf{w}, b, \mathbf{x})$, we sample and evaluate $N = D^2$ neighboring perturbations of $\boldsymbol{\delta} = \tilde{R}\mathbf{w}/\|\mathbf{w}\|$ using the Neighbor function (Algorithm 3 in Appendix A).

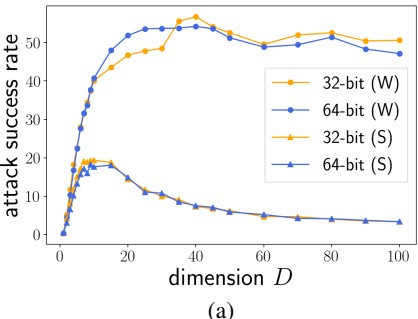 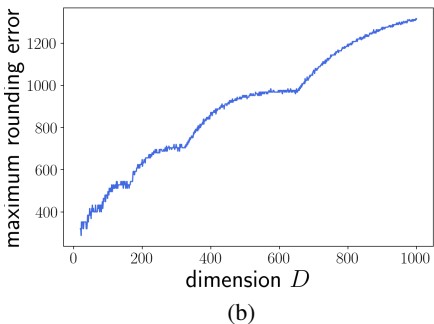

(a)            (b)

Figure 2: (a) Rounding search attack success rates against a random binary linear classifier in both weak (W) and strong (S) threat models (Section 4.1). For each dimension $D$, we report the percentage of 10 000 randomly initialized models for which we can successfully find an adversarial example within certified radius $\tilde{R}$ for a random instance $\mathbf{x}$ drawn from $[-1, 1]^D$. Since the attacks are against an exact certified radius, the baseline attack rate should be $0\%$ in both weak and strong threat models. Model weights $\mathbf{w}$ and biases $b$ are randomly initialized with $\mathbf{w} \in [-1, 1]^D$, $b \in [-1, 1]$. All values and computation is done using either 32-bit or 64-bit floating points. (b) Maximum rounding error in the calculation of the certified radius $\tilde{R}$ on a sample $\mathbf{x}$ with each $x_i = 3.3 \times 10^9$, for the linear model with $w_i = 3.3 \times 10^{-9}$, $b = 3.3 \times 10^9$, where $i \in [1, D]$ and $D \in [20, 1000]$.

Results are shown in Figure 2(a). With higher dimension, our attack success rate first increases and then flattens around $50\%$ in the weak threat model, and around $5\%$ in the strong threat model. (we investigate the flattening phenomenon in Appendix E). With higher dimension more arithmetic operations are done in computing $\|\boldsymbol{\delta}'\|$ and $\tilde{R}$, which results in accumulation of rounding errors. Figure 2(b) further shows this influence of $D$ on the rounding error, which can be accumulated to the magnitude of $10^3$ with increasing $D$. In summary, with the increasing rounding error, a greater leeway is left between the real certified radius $R$ and the computed certified radius $\tilde{R}$ for our method to exploit, so the attack success rate increases.

The success rates are lower in the strong threat model than in the weak threat model. This is expected, as the leeway (*i.e.*, $\tilde{R} - \overline{\|\boldsymbol{\delta}\|}$) exploited by our attack in the conservative strong model is likely much smaller than that (*i.e.*, $\tilde{R} - \|\boldsymbol{\delta}\|$) in the weak model.

## 4.2 LINEAR SVM

In this section, we evaluate our attack on linear SVM trained with the MNIST dataset. MNIST LeCun et al. (2010) contains images of hand-written digits where each image has 784 attributes and each attribute is a pixel intensity within the domain $[0, 255]$. We used $\approx 12\,000$ images for training, and $\approx 2\,000$ images for validation and evaluation of our attacks, for each combination of the labels $i, j \in \{0, \ldots, 9\}$. We trained 45 models for each combination of distinct labels $i, j \in \{0, \ldots, 9\}$ of the MNIST dataset. Validation accuracies range between $91\%$ and $99\%$ for linear SVM.

We then try to find an adversarial image with respect to each image in the test dataset. Our attack samples $N = 5\,000$ neighbors of $\boldsymbol{\delta} = \tilde{R}\mathbf{w}/\|\mathbf{w}\|$ using Algorithm 3 of Appendix A.

In the weak threat model, we observe non-zero attack success rates for 44/45 models (full results appear in Table 1 of Appendix B), and our attacks can have success rates up to $23.24\%$. In the strong threat model, we observe non-zero attack success rates for 11/45 models (full results appear in Table 2 of Appendix B), and our attacks can have success rates up to $0.16\%$. Recall that the baseline success rate should always be $0\%$. We demonstrate a weak model example of original and adversarial images together with their perturbation and certified radius information in Figure 3 of Appendix B.

### 4.3 CERTIFICATION FOR NEURAL NETS

We now turn our attention to neural network verification mechanisms. In this section we consider neural networks with ReLU activations and rely on their linear approximations. Given a radius $\tilde{R}$, a neural network $F$ and an input $\mathbf{x}$, these mechanisms either certify $\tilde{R}$ or not. Hence, in order to find a tight certified radius for a given model, one can perform a binary search to check multiple radii and call those verifiers multiple times. We avoid the binary search to find a certified radius $\tilde{R}$ by first finding an adversarial example $\mathbf{x}'$ via ReluPGD (Section 3.1 and Algorithm 2 of Appendix A) and then try to verify the perturbation norms (*i.e.*, $\|\mathbf{x}' - \mathbf{x}\|$) of those adversarial examples using the complete verifiers. We set ReluPGD to time out after 15 minutes.

**Certification with $\beta$-CROWN.** $\beta$-CROWN Wang et al. (2021) guarantees sound but not complete robustness certification. That is, it provides a lower bound on the radius and it is possible that a tighter radius may exist. We use the $\beta$-CROWN verifier Wang et al. (2021) in the $\ell_2$ metric, to verify a 3-layer neural network binary classifier with 1 node in the hidden layer. All model weights and biases in the hidden layers of this classifier are trained to be positive, so the perturbation direction is always $\boldsymbol{\nu} = \mathbf{w}_t^T - \mathbf{w}_l^T$. The classifier has validation accuracy 99.67%. We use ReluPGD, with step size $s = 1 \times 10^{-5}$, to incrementally add perturbation in direction $\boldsymbol{\nu}$ to image $\mathbf{x}$, until its prediction is flipped, and we get $\mathbf{x}'$. Then we use $\beta$-CROWN to verify the image with respect to $\|\boldsymbol{\delta}\| = \|\mathbf{x}' - \mathbf{x}\|$. If the verification succeeds, we have a successful attack. ReluPGD times out on 30 out of 2 108 images. When the attack does not time out, it takes $\approx 30$ seconds. A call to a verifier takes $\approx 1$ second. We conduct our attack on all MNIST test images labeled 0 or 1. We find adversarial images for 2 078 images, and $\beta$-CROWN erroneously verifies 53 of them. Our attack success rate is 2.6%.

**Certification with MIP solver.** We now consider another method that provides conservative verification via mixed-integer programming (MIP). We use the implementation from Wang et al. (2021), which uses the Gurobi MIP solver Gurobi Optimization, LLC (2022) for neural network verification. We verify two 3-layer neural network multiclass classifiers with 100 nodes in their hidden layer. For the first classifier, the weights and biases in the hidden layers are all trained to be positive, so the perturbation direction is $\boldsymbol{\nu} = \mathbf{w}_t^T - \mathbf{w}_l^T$. The second classifier is trained without constraints on its weights and represents a regular network without artefacts. The validation accuracies for the first and second classifiers are 84.14%, and 96.73%, respectively.

We use ReluPGD to attack the two classifiers on all images of the MNIST test dataset, with step size $s = 1 \times 10^{-5}$. We found adversarial images against 8 406 images for the first classifier, and adversarial images against 9 671 images for the second classifier. ReluPGD times out on only 8 and 2 images for the first and second classifier, respectively. We then use MIP to verify each image with respect to their adversarial image's perturbation norm $\|\boldsymbol{\delta}\|$. Each attack takes $\approx 30$ seconds and each verification takes $\approx 10$ seconds. MIP successfully verified 5 108 out of 8 406 successfully attacked images for the first classifier, and verified 1 531 out of 9 671 successfully attacked images. That is, the attack success rate is 60.76% on an artificially trained network (where ReLUs are all activated) and 15.83% on the second classifier trained without artefacts.

**Certification with randomized smoothing.** We also attack approximate certification methods based on randomized smoothing Cohen et al. (2019). Recall that guarantees of randomized smoothing are probabilistic with a failure probability $\alpha$. Nevertheless we report a success rate up to 21.11% for our rounding search attacks on MNIST with $\alpha = 0.1\%$. We refer readers to Appendix C for details.

## 5 MITIGATION: CERTIFICATION WITH ROUNDED INTERVAL ARITHMETIC

Our attack results demonstrate that floating-point rounding invalidates the soundness claims of a wide range of certification implementations for a variety of common models. How might such rounding errors in certification calculations be mitigated, for both of our threat models?

Rounding errors violating certifications are sometimes small. For example, the rounding error for the certified radius of the first MNIST image of Figure 3 (Appendix A) is in the 13th decimal place. One's first intuition may be to adopt slightly more conservative radii (*e.g.*, using $\tilde{R} - \gamma$ for some positive constant $\gamma \ll 1$). Unfortunately, such radii are not in general sound, and attacks against

$\tilde{R} - \gamma$ are still possible. For example, as we show in Section 4.1, it is easy to construct a linear classifier and find adversarial examples against it within $\tilde{R} - 0.1$.

We outline a mitigation applying rounded interval arithmetic Higham (2002) to certified robustness. Interval arithmetic replaces every numerical value with an interval. Interval operators exist for elementary arithmetic operations, serving as useful building blocks for more complex computations with bounded rounding errors (Definition 2 of Appendix D). We have re-framed existing results from numerical analysis in the language of sound floating-point computation (Lemma 1 of Appendix D).

**Theorem 1** *Consider a classifier $f$, floating-point instance $\mathbf{x}$, and a certification mechanism $R(f, \mathbf{x})$ that is sound when employing real arithmetic. If $R(f, \cdot)$ can be computed by a composition of real-valued operators $\psi_1, \ldots, \psi_L$ with sound floating-point extensions $\phi_1, \ldots, \phi_L$, then the following certification mechanism $\underline{R}(f, \mathbf{x})$ is sound with floating-point arithmetic: run the compositions of $\phi_1, \ldots, \phi_L$ on (coordinate-wise) intervals $[\mathbf{x}, \mathbf{x}], [f(\mathbf{x}), f(\mathbf{x})]$ to obtain $[\underline{R}, \overline{R}]$; return $\underline{R}$.*

The proof of Theorem 1 appears in Appendix D.1. We offer an example application of this mitigation theorem on linear classifiers. We use the PyInterval library Taschini (2008) that performs rounded interval arithmetic to compute sound $\underline{R}$ for linear classifiers Cohen et al. (2019). Our attack success rates for randomly initialized linear classifiers (Section 4.1) drop to 0% for all dimensions in both weak and strong threat models. In sum, our theoretical and empirical results provide support for mitigating attacks against exact robustness certifications Cohen et al. (2019).

# 6 RELATED WORK

Several works have explored the influence of floating-point representation on guarantees of verified neural networks. For example, verifiers designed for floating-point representation have been shown to not necessarily work for quantized neural networks Giacobbe et al. (2020); Henzinger et al. (2021).

The closest to our work is the independent work by Jia & Rinard (2021) who also exploit rounding errors to discover violations of network robustness certifications. Our work differs from Jia & Rinard (2021) on the adversarial examples we find. As we show in Section 4.3, we are able to find an adversarial example $\mathbf{x}'$ for unaltered natural image $\mathbf{x}$ from test data, within that image $\mathbf{x}$'s certified radius. The work by Jia and Rinard, instead, does not find certification-violating adversarial examples of test instances. It finds perturbed inputs $\mathbf{x}'_0$ of synthetic inputs $\mathbf{x}_0$, that violate certifications of $\mathbf{x}_0$. In particular, they adjust brightness of a natural test image $\mathbf{x}$ to produce a $\mathbf{x}_0$. That is, their attack point $\mathbf{x}'_0$ is *outside the certified radius of* $\mathbf{x}$. Hence, our attack can be seen as a stronger attack that is possible due to a novel attack methodology based on accurate perturbation directions.

Research in the area of numerical analysis has proposed approaches to address the limitations of floating-point rounding, with a focus on measuring the stability of calculations. Proposed approaches include replacing floating-point arithmetic with interval arithmetic Jaulin et al. (2001) or affine arithmetic De Figueiredo & Stolfi (2004). Both account for rounding errors and return an interval that contains the correct result. Our work is the first to suggest that modern systems implementing these approaches could be of use to certified robustness implementations. We adopt interval arithmetic with the implementation PyInterval Taschini (2008) in the calculation of robustness certification.

# 7 CONCLUSION

Certified robustness has been proposed as a defense against adversarial examples. In this work we have shown that guarantees of several certification mechanisms do not hold in practice since they rely on real numbers that are approximated on modern computers. Hence, computation on floating-point numbers—used to represent real numbers—can overestimate certification guarantees due to rounding. We propose and evaluate a rounding search method that finds adversarial inputs on linear classifiers and verified neural networks within their certified radii—violating their certification guarantees. We propose rounded interval arithmetic as the mitigation, by accounting for the rounding errors involved in the computation of certification guarantees. We conclude that if certified robustness is to be used for security-critical applications, their guarantees and implementations need to account for limitations of modern computing architecture.

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
