# A   ALGORITHM PSEUDOCODE

The pseudocode of the Linearized ReLU Network Gradients and the Linearized Projected Gradient Descent for ReLU NNs from Section 3.1, and Floating-point Neighbors Search algorithm from Section 3.2 appear in Algorithm 1, Algorithm 2, and Algorithm 3 respectively.

In Algorithm 3, beforeFP($v$) returns the first floating-point value before $v$, and afterFP($v$) returns the first floating-point value after $v$. In the experiments, we set $p = 2$ for Algorithm 3.

---

**Algorithm 1:** LinApproxPerturbDir: Linearized ReLU Network Gradients

---

**Input:** input to be perturbed $\mathbf{x}$; the neural network model $F(\mathbf{x}) = F_n \circ F_{n-1} \circ \cdots \circ F_1(\mathbf{x})$,
$\quad$ $F_i(\mathbf{x}) = \mathsf{ReLU}(\boldsymbol{\theta}_i^T \mathbf{x} + \hat{\boldsymbol{\theta}}_i)$; current label $l$; adversarial target label $t$.
**Output:** $\boldsymbol{\nu}$, a perturbation direction.
**Function** LinApproxPerturbDir ($\mathbf{x}$, $F$, $l$, $t$):
$\quad$ $\mathbf{h}_1 \leftarrow \boldsymbol{\theta}_1^T \mathbf{x} + \hat{\boldsymbol{\theta}}_1$
$\quad$ **for** $i \leftarrow 2$ **to** $n$ **do**
$\quad\quad$ $\mathbf{m}_i \leftarrow \mathbb{1}_{[\mathbf{h}_{i-1}>0]}$ $\quad \triangleright$ elementwise thresholding
$\quad\quad$ $\boldsymbol{\tau}_i^T \leftarrow \boldsymbol{\theta}_i^T \odot \mathbf{1}\mathbf{m}_i^T$ $\quad \triangleright$ Hadamard product; $\mathbf{1}$ is a column vector of
$\quad\quad\quad$ 1s
$\quad\quad$ $\mathbf{z}_{i-1} \leftarrow \mathsf{ReLU}(\mathbf{h}_{i-1})$
$\quad\quad$ $\mathbf{h}_i \leftarrow \boldsymbol{\theta}_i^T \mathbf{z}_{i-1} + \hat{\boldsymbol{\theta}}_i$
$\quad$ **end**
$\quad$ $\boldsymbol{\tau}^T \leftarrow \boldsymbol{\tau}_n^T \boldsymbol{\tau}_{n-1}^T \cdots \boldsymbol{\tau}_2^T \boldsymbol{\theta}_1^T$ $\quad \triangleright$ weights of $F'$
$\quad$ $\boldsymbol{\nu} \leftarrow \boldsymbol{\tau}^T[t] - \boldsymbol{\tau}^T[l]$
$\quad$ **return** $\boldsymbol{\nu}$
**End Function**

---

**Algorithm 2:** ReluPGD: Linearized Projected Gradient Descent for ReLU NNs

---

**Input:** input to be perturbed $\mathbf{x}$; the neural network model $F(\mathbf{x}) = F_n \circ F_{n-1} \circ \cdots \circ F_1$,
$\quad$ $F_i(\mathbf{x}) = \mathsf{ReLU}(\boldsymbol{\theta}_i^T \mathbf{x} + \hat{\boldsymbol{\theta}}_i)$; current label $l$; adversarial target label $t$; step size $s$; input
$\quad$ domain $[V_{\mathsf{min}}, V_{\mathsf{max}}]$.
**Output:** $\boldsymbol{\delta}$, adversarial perturbation.
**Function** ReluPGD ($\mathbf{x}, F, l, t, s, V_{\mathsf{min}}, V_{\mathsf{max}}$):
$\quad$ $\mathbf{x}' \leftarrow \mathbf{x}$ $\quad \triangleright$ initial adversarial example
$\quad$ **do**
$\quad\quad$ $\boldsymbol{\nu} \leftarrow \mathsf{LinApproxPerturbDir}(\mathbf{x}', F, l, t)$
$\quad\quad$ $\boldsymbol{\delta} \leftarrow \frac{s}{\|\boldsymbol{\nu}\|} \boldsymbol{\nu}$
$\quad\quad$ $\mathbf{x}' \leftarrow \mathbf{x}' + \boldsymbol{\delta}$
$\quad\quad$ $\mathbf{x}' \leftarrow \mathsf{clip}(\mathbf{x}', V_{\mathsf{min}}, V_{\mathsf{max}})$
$\quad$ **while** $F(\mathbf{x}') \neq t$ $\quad \triangleright$ or till timeout
$\quad$ **return** $\boldsymbol{\delta} \leftarrow \mathbf{x}' - \mathbf{x}$
**End Function**

---

**Algorithm 3:** Neighbor: FP Neighbors Search

---

**Input:** perturbation seed vector $\boldsymbol{\delta} = [\delta_1, \delta_2, \ldots, \delta_D]$;
$p$: number of signed neighbor values to sample from for each dimension;
$N$: number of sampled neighbors of $\boldsymbol{\delta}$ to return;
**Output:** $\boldsymbol{\delta}$-neighb, $N$ neighbors of $\boldsymbol{\delta}$
**Function** Neighbor ($\boldsymbol{\delta}$, $N$, $n$):
$\quad$ candidates $\leftarrow [(2p + 1) \times D]$
$\quad$ $\triangleright$ Continues next page

```
for i ← 1 to D do
    candidates[1][i] ← δ_i
    r_i ← δ_i
    l_i ← δ_i
    j ← 2
    while j < 2p + 1 do
        r_i ← afterFP(r_i)
        l_i ← beforeFP(l_i)
        candidates[j][i] ← r_i
        candidates[j + 1][i] ← l_i
        j ← j + 2
    end
end
δ-neighb ← sample(candidates, N) ▷ Construct N neighbor points:  for
    each point δ', randomly sample component δ'_i from candidates[·][i],
    such that δ–neighb contains no duplicates nor copies of δ.
return δ-neighb
End Function
```

## B    ATTACK RESULTS ON MNIST FOR LINEAR SVM

The results of rounding search attacks on linear SVM from Section 4.2 for each combination of the distinct labels $i, j \in \{0, \dots, 9\}$ of the MNIST dataset are listed in Table 1 for the weak model and Table 2 for the strong model. An example of original and adversarial images together with their perturbation and certified radius information based on linear SVM attack is presented in Figure 3.

| labels | 1 | 2 | 3 | 4 | 5 | 6 | 7 | 8 | 9 |
|---|---|---|---|---|---|---|---|---|---|
| 0 | 11.06% | 5.12% | 3.62% | 2.65% | 0.53% | 2.58% | 6.18% | 3.43% | 3.12% |
| 1 | | 5.77% | 5.64% | 6.8% | 23.24% | 0.38% | 16.04% | 4.46% | 6.06% |
| 2 | | | 1.08% | 0.3% | 1.3% | - | 1.12% | 1.0% | 1.03% |
| 3 | | | | 0.1% | 1.21% | 5.18% | 4.47% | 1.31% | 0.25% |
| 4 | | | | | 1.92% | 0.05% | 4.08% | 1.84% | 0.4% |
| 5 | | | | | | 2.16% | 5.78% | 0.05% | 0.58% |
| 6 | | | | | | | 2.92% | 10.09% | 0.05% |
| 7 | | | | | | | | 2.3% | 1.57% |
| 8 | | | | | | | | | 1.06% |

Table 1: The success rates of our rounding search attack against linear SVM models on the MNIST dataset in the weak threat model (Section 4.2). An experiment is successful if adversarial perturbation $\|\boldsymbol{\delta}\|$ is less than or equal to certified radius $\tilde{R} = |\mathbf{w}^T \mathbf{x} + b| / \|\mathbf{w}\|$ computed with finite-precision floating-point arithmetic (*i.e.*, $\|\boldsymbol{\delta}\| \leq \tilde{R}$). 45 linear SVM models have been trained and attacked for each combination of labels respectively. A cell in row $i$ and column $j$ reports the attack success rate for classes original $i$ and target $j$, plus the attack success rate for classes original $j$ and target $i$. For example, for an SVM model, we can find adversarial perturbations within certified radius for 11.06% of all images labelled 1 (with the model classifying them as 0) or images labelled 0 (with the model classifying them as 1). We use "-" to denote models where rounding search did not find an adversarial example.

## C    ATTACK RESULTS ON MNIST FOR APPROXIMATE CERTIFICATION

**Randomized smoothing for approximate certification.**    Researchers have studied approximate certification of non-linear models such as neural networks, under the $\ell_2$ norm Lecuyer et al. (2019); Li et al. (2019); Cohen et al. (2019). In this work, we consider one such approach of Cohen *et al.* Cohen et al. (2019) from which many subsequent results derive. Sufficiently stable model predictions lead

| labels | 1 | 2 | 3 | 4 | 5 | 6 | 7 | 8 | 9 |
|---|---|---|---|---|---|---|---|---|---|
| 0 | - | - | - | - | - | - | - | - | 0.05% |
| 1 | | - | 0.05% | 0.05% | - | - | - | - | - |
| 2 | | | - | - | - | - | - | - | - |
| 3 | | | | - | 0.16% | 0.05% | - | 0.05% | - |
| 4 | | | | | - | 0.05% | - | 0.05% | - |
| 5 | | | | | | - | - | 0.05% | 0.16% |
| 6 | | | | | | | - | - | 0.05% |
| 7 | | | | | | | | - | - |
| 8 | | | | | | | | | - |

Table 2: The success rates of our rounding search attack against linear SVM models on the MNIST dataset in the strong threat model (Section 4.2). An experiment is successful if the upper bound of adversarial perturbation $\overline{\|\boldsymbol{\delta}\|}$ is less than or equal to certified radius $\tilde{R} = |\mathbf{w}^T\mathbf{x} + b|/\|\mathbf{w}\|$ computed with finite-precision floating-point arithmetic (*i.e.*, $\overline{\|\boldsymbol{\delta}\|} \leq \tilde{R}$). 45 linear SVM models have been trained and attacked for each combination of labels respectively. A cell in row $i$ and column $j$ reports the attack success rate for classes original $i$ and target $j$, plus the attack success rate for classes original $j$ and target $i$. We use "-" to denote models where rounding search did not find an adversarial example.

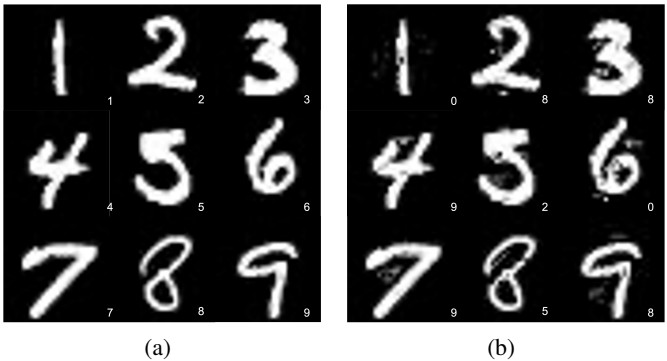

(a)  (b)

Figure 3: (a) Original images from the MNIST dataset. (b) Corresponding adversarial images in the weak threat model, with perturbations within the exact certified radius (*i.e.*, $\|\boldsymbol{\delta}\| \leq \tilde{R}$) but the linear SVM model misclassifies them. For example, the first top-left image in (a) has certified radius of $\tilde{R} = 333.6087764918925$, and is classified as 1, while the corresponding adversarial image in (b) has perturbation $\|\boldsymbol{\delta}\| = 333.6087764918924$, and is classified as 0. Labels at the bottom right of each image are the classifications of the linear SVM model (Section 4.2).

to certifiable radii; therefore we seek to stabilize the outputs of the base classifier $f$ by forming a smoothed classifier $g$ as follows. To input $\mathbf{x} \in \mathbb{R}^D$ add isotropic Gaussian noise, then apply $f$. The input distribution induces a distribution over predictions, and outputs the most likely class. That is, for $\boldsymbol{\epsilon} \sim \mathcal{N}(0, \sigma_P^2 I^2)$, $g(\mathbf{x}) = \arg\max_{k\in[K]} \Pr\left(f(\mathbf{x} + \boldsymbol{\epsilon}) = k\right)$.

Consider lower (upper) bounds on the winning $k^\star$ classification's probability score, $\Pr(f(\mathbf{x} + \boldsymbol{\epsilon}) = k^\star) \geq \underline{p_A} \geq \overline{p_B} \geq \max_{k\in[K]\setminus\{k^\star\}} \Pr(f(\mathbf{x} + \boldsymbol{\epsilon} = k))$. Then Cohen *et al.* Cohen et al. (2019) prove that radius $R = 0.5\sigma_P(\Phi^{-1}(\underline{p_A}) - \Phi^{-1}(\overline{p_B}))$ is a sound certification for smoothed classification $g(\mathbf{x})$, under $\mathbb{R}$ arithmetic. However the floating-point calculation of a corresponding $\tilde{R}$ is likely to experience some degree of rounding, potentially sufficient to erroneously certify some $\tilde{R} > R$.

**Setting** Our experiments on approximate certification attack a 3-layer ReLU network with 100 nodes in the hidden layer trained on the MNIST LeCun et al. (2010) dataset with labels 0 and 1, which has validation accuracy 99.95%. Pixel values of all images are scaled to $[0, 1]$.

**Prediction and certified radius estimation.** We adopt the same prediction procedures as Li et al. (2019). Given an instance $\mathbf{x}$, the smoothed classifier $g$ runs the base classifier $f$ on $M$ noise-corrupted instances of $\mathbf{x}$, and returns the top class $k_A$ that has been predicted by $f$. The estimation of the certified radius $\tilde{R}$ for a smoothed classifier $g$ is based on the probability distribution of the winner and runner-up classes (*i.e.*, $p_A$ and $p_B$). $p_A$ and $p_B$ are estimated via interval estimation Brown et al. (2001), with confidence $1 - \alpha$. As recommended in these papers, we set $\alpha = 0.1\%$, Gaussian prediction noise scale $\sigma_P \in \{1.0, 3.0, 5.0, 7.0\}$, and let the number of Monte Carlo samples $M$ range in $\{100, 10000\}$.

**Attack.** We conduct our attack as follows. Given $\mathbf{x}$ and $\tilde{R}$ as returned by $g$, we first use ReluPGD (maximum runtime is set as 15 minutes) to find an adversarial perturbation $\boldsymbol{\delta}$ against the base classifier $f$, then use Neighbor to find $N = 1000$ neighbors $\boldsymbol{\delta}'$ of $\boldsymbol{\delta}$. We evaluate all generated adversarial examples $\mathbf{x} + \boldsymbol{\delta}'$ using the robust classifier $g$. For an image $\mathbf{x}$, if any one $\mathbf{x} + \boldsymbol{\delta}'$ of the $N = 1000$ adversarial images flips the classification of the model with $\|\boldsymbol{\delta}'\| \leq \tilde{R}$, we have a successful attack against the approximate certification on this image. We chose $f$ in the process of perturbation generation as we need a concrete ReLU network with clear weights. We conduct attacks on 2 115 images of the MNIST dataset with labels 0 and 1, and have success rates up to 21.11% against approximate certification. Detailed results are listed in Table 3.

| $\sigma_P$ | $M$ | $N_A$ | $N_V$ | success rate |
|---|---|---|---|---|
| 1.0 | 100 | 5 | 2111 | 0.24% |
| | 10000 | 1 | 2115 | 0.05% |
| 3.0 | 100 | 182 | 2066 | 8.81% |
| | 10000 | 8 | 2113 | 0.38% |
| 5.0 | 100 | 273 | 1606 | 17.00% |
| | 10000 | 54 | 2109 | 2.56% |
| 7.0 | 100 | 220 | 1042 | 21.11% |
| | 10000 | 99 | 2066 | 4.79% |

Table 3: The attack success rates of our rounding search within the certified radius $\tilde{R}$ ($\alpha = 0.1\%$) Cohen et al. (2019). The number of Monte Carlo samples $M$ used in certified radius estimation ranges in $\{100, 10000\}$. Prediction noise scale $\sigma_P \in \{1.0, 3.0, 5.0, 7.0\}$. We randomly sample and evaluate $N = 1000$ neighbors $\boldsymbol{\delta}'$ of $\boldsymbol{\delta}$ in the search area for each image. $N_A$ and $N_V$ denote the number of examples that have been successfully attacked and verified respectively.

# D MITIGATION DEFINITIONS

As outlined in Section 5, interval arithmetic is an approach to bounding approximations in numerical analysis. It involves replacing each (possibly) approximate floating-point scalar with an interval with floating-point end-points that are guaranteed to bound the scalar. Our goal is to enable any computation on floating-points to be possible on interval representations, such that this 'guaranteed bounding' property on input data, parameters, or intermediate computations, is invariant to further computation.

It is first necessary to extend the standard arithmetic operators to interval arithmetic. The following definition demonstrates this process assuming arithmetic on $\mathbb{R}$: provided two target values $a, b$ are contained in intervals $[a_1, a_2], [b_1, b_2]$ to begin with, then the presented operators are guaranteed to *maintain* this property on basic arithmetic operations. For example, $a + b \in ([a_1, a_2] + [b_1, b_2])$, where the operator '+' on reals is overloaded to real intervals.

**Definition 2** *For real intervals $[a_1, a_2], [b_1, b_2]$, define the following interval operators for elementary arithmetic:*

- *Addition $[a_1, a_2] + [b_1, b_2]$ is defined as $[a_1 + b_1, a_2 + b_2]$.*

- *Subtraction $[a_1, a_2] - [b_1, b_2]$ is defined as $[a_1 - b_2, a_2 - b_1]$.*

- *Multiplication $[a_1, a_2] * [b_1, b_2]$ is defined as $[\min\{a_1 b_1, a_2 b_1, a_1 b_2, a_2 b_2\}, \max\{a_1 b_1, a_2 b_1, a_1 b_2, a_2 b_2\}]$.*

- *Division $[a_1, a_2]/[b_1, b_2]$ is defined as $[a_1, b_2] \times 1/[b_1, b_2]$ where by cases:*

$$\frac{1}{[b_1, b_2]} = \begin{cases} \left[\frac{1}{b_2}, \frac{1}{b_1}\right], & \text{if } 0 \notin [b_1, b_2] \\ \left[-\infty, \frac{1}{b_1}\right] \cup \left[\frac{1}{b_2}, \infty\right], & \text{otherwise} \end{cases}$$

$$\frac{1}{[b_1, 0]} = \left[-\infty, \frac{1}{b_1}\right], \quad \frac{1}{[0, b_2]} = \left[\frac{1}{b_2}, \infty\right]$$

*Rounded* interval arithmetic Higham (2002) further employs floating-point rounding when implementing the arithmetic operators, to achieve these sound floating-point extensions.

**Lemma 1** *Consider the interval arithmetic operators in Definition 2 with the resulting lower (upper) interval limits computed using IEEE754 floating-point arithmetic with rounding down (up), then the resulting* rounded interval arithmetic operators *are sound floating-point extensions.*

That is, given a collection of floating-point intervals representing a collection of corresponding reals, rounded extension operators produce floating-point intervals that are guaranteed to contain the result of corresponding (base, unextended) arithmetic on the given collection of reals.

By representing constants as singleton intervals, rounded interval arithmetic computes floating-point bounds on real-valued results. Applying Definition 2 and Lemma 1, we can compute a sound floating-point interval for the rational $1/3$:

$$\frac{[1,1]}{[3,3]} = [1,1] * \frac{1}{[3,3]} = [1,1] * \left[\frac{1}{3}, \frac{1}{3}\right] = \left[\left\lfloor\frac{1}{3}\right\rfloor, \left\lceil\frac{1}{3}\right\rceil\right] = [0.33\ldots33, 0.33\ldots337]$$

Beyond rounded interval arithmetic operators, libraries such as PyInterval offer rounded interval implementations of standard algebraic functions (*e.g.*, the square root) and transcendental functions (*e.g.*, the exponential, logarithm, trigonometric, and hyperbolic functions) using Newton-Raphson approximation that itself applies the above basic operators. Such functionality enables application of rounded interval arithmetic to general machine learning models and their certifications.

### D.1 PROOF OF THEOREM 1

The result follows by strong induction on the levels of composition implementing $\underline{R}(f, x)$, with repeated application of Lemma 1. The induction hypothesis is that up to $l \leq L$ levels of composition of interval operators produces intervals that contain the result of the corresponding real operators. The base case comes from the (coordinate-wise) intervals $[\mathbf{x}, \mathbf{x}], [f(\mathbf{x}), f(\mathbf{x})]$ containing coordinates of $\mathbf{x}$ and $f(\mathbf{x})$; the inductive step follows from repeated application of sound floating-point extensions.

## E  REVISITING ATTACKS

We now revisit our Section 4.1 experiment on randomized binary linear classifiers. Figure 2(a) observes an intriguing behavior of fast initial improvement of attack success rate to 50% followed by asymptoting in the weak threat model, for both 32-bit and 64-bit representations.

To explore why our success rate flattens, we conduct an experiment in the weak threat model with 64-bit representation. We use the PyInterval library Taschini (2008) to calculate the lower and upper bounds, $\underline{R}$ and $\overline{R}$, of certified radius $R$. Then we conduct a binary search in $[\underline{R}, \overline{R}]$ to find the maximum certified radius $\hat{R}$, within which there are no adversarial examples (Theorem 1). We take $\hat{R}$ as our best approximation to $R$, and use it to estimate the rounding error of $\tilde{R}$ (*i.e.*, $\hat{R} - \tilde{R}$).

As in Figure 4(a), our attack against linear models exploits rounding errors in the magnitude of $10^{-15}$. We find that the rate that $\tilde{R}$ is overestimated ($\tilde{R} > \hat{R}$) flattens around 50% (Figure 4(b)). Recall that our attack works when $\tilde{R}$ is overestimated, so there is leeway for our attack to exploit. Therefore, our success rate flattens around 50% because that is the maximum rate that $\tilde{R}$ is overestimated.

Why does the overestimated rate flatten around 50%? One intuition is that rounding is effectively random: there is a 50% chance for $\tilde{R}$ to be overestimated (rounded up), and another 50% chance for

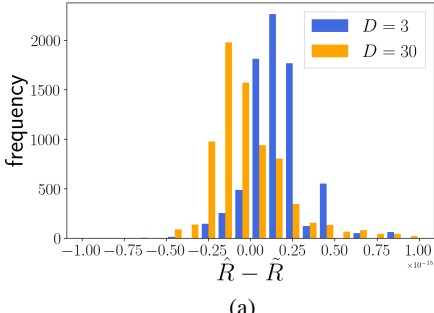 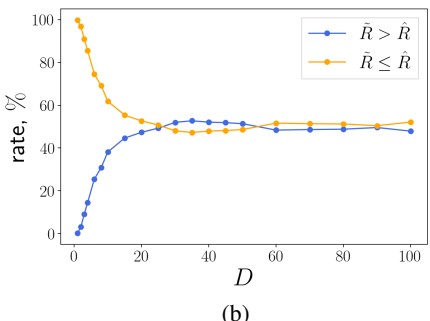

(a)                                        (b)

Figure 4: Exploration of flattening success rate phenomenon in Section 4.1. The experiment is done in the weak threat model with 64-bit floating-point representation. $\tilde{R}$ is the estimated certified radius using IEEE 754 floating-point arithmetic (with rounding errors), $\hat{R}$ is the certified radius found via binary search within $[\underline{R}, \overline{R}]$, $\underline{R}$ and $\overline{R}$ are the lower and upper bounds of the real certified radius $R$ estimated using interval arithmetic Taschini (2008). $\hat{R}$ is our best approximation to $R$, within which there are no robustness violations. (a) Deviations ($\hat{R} - \tilde{R}$) between $\tilde{R}$ and $\hat{R}$ over 10 000 trials for $D \in \{3, 30\}$. (b) For each dimension $D \in [1, 100]$, we plot percentage of 10 000 trials for which $\hat{R} > \tilde{R}$ and $\hat{R} \leq \tilde{R}$. Our attacks may work when $\hat{R} < \tilde{R}$, that is, the certified radius is overestimated.

$\tilde{R}$ to be underestimated (rounded down). Hence, no matter in 32-bit or 64-bit representation, our attack success rates flatten around 50%.

## F    BROADER IMPACT

Our attacks could potentially be used to find violations of robustness guarantees in system implementations, by exploiting floating-point vulnerabilities in those implementations. Note that to date, we are not aware of any certified robustness implementations yet in deployment. However since certifications could be deployed in the near future, this work serves an important role in highlighting this new vulnerability, and in proposing an effective mitigation against it.

## G    LIMITATIONS

We identify the following limitations that could be addressed as future work. First, our attacks against methods based on randomized smoothing did not study the relationship between the attack success rate and the soundness probability of these methods due to their probabilistic nature. A possible future direction would be to study how the two interact. Second, we showed how to adopt our mitigation based on interval arithmetic generally, and demonstrate this mitigation for linear models and exact certifications. Detailed exploration of embedding these mitigations in other verifiers and models is left as future work.