# OpenReview forum: "Getting a-Round Guarantees: Floating-Point Attacks on Certified Robustness"
_ICLR.cc/2024/Conference — ICLR 2024 Conference Withdrawn Submission_

### Official Review · Reviewer_GFuC · 2023-10-16

**Soundness:** 2 fair
**Presentation:** 2 fair
**Contribution:** 1 poor
**Rating:** 3
**Confidence:** 4

**Summary:**

The paper investigates the floating-point soundness of adversarial robustness certificates in two settings: First, a "weak threat model" where both the perturbation norm and the certified radius are computed using floating-point arithmetic and second, a "strong threat model" where a floating-point sound upper-bound on the perturbation norm is used instead. In the strong setting, the paper shows soundness violations for both random linear models and SVMs trained on the binary distinction of MNIST digits, where the (up to floating-point soundness) exact certified radius can be computed trivially. In the weak setting, the paper additionally shows floating point violations for small (positive) neural networks, analyzed with popular methods. Their attack is based on a PGD variant combined with a random search over neighboring floating-point values. Finally, the paper proposes to address this issue by using interval arithmetic to compute the robustness certificates.

**Strengths:**

## Strengths
* The tackled issue of (certified) adversarial robustness is of high importance.
* The paper is mostly easy to understand.
* To the best of my knowledge, this paper is the first to demonstrate floating point attacks around test-set images on unperturbed networks.
* While simple, the random floating point neighbour search (Alg. 3) seems effective.

**Weaknesses:**

## Weaknesses
* The proposed ReluPGD (Algorithm 1 and 2) seems to simply recover standard PGD with $\ell_2$ projections, which the authors do not acknowledge.
* The literature review is missing key works such as Singh et al. (2018 and 2019), which propose floating-point sound neural network verifiers leveraging interval arithmetic, the exact approach proposed as novel by this paper. Similarly although less crucial, related work on floating-point sound Randomised Smoothing is missing (Voráček and Hein 2022).
* Experimental details are missing at many points (e.g. 32 vs 64-bit floating-point precision, Gurobi tolerances and precisions) that make it not only hard to reproduce results but even to assess their significance.
* In many settings, only the "weak threat model" is effective, which mostly seems to attack the floating-point soundness of the norm computation (see Figure 2a) and not the network verifier. As success rates for non-random linear classifiers are less than 0.2 % in the strong setting (with no success for non-linear classifiers), this should be communicated clearly.
* The claim of verifying "Neural Nets" with "conservative" methods is misleading given that 2 of the 3 considered neural nets are actually linear (positive inputs and positive parameters ensure all ReLUs become identity functions), in which case verification is exact and the last one is only analyzed with MILP, which for a network of such small size should not only theoretically but also practically be exact.

**References**
* Voráček, Václav, and Matthias Hein. "Sound randomized smoothing in floating-point arithmetics." arXiv 2022
* Singh, Gagandeep, et al. "An abstract domain for certifying neural networks."  POPL 2019
* Singh, Gagandeep, et al. "Fast and effective robustness certification." NeurIPS 2018

**Questions:**

### Questions
1) It seems that the gradients of the linearized network are identical to those of the standard model. Thus Alg. 1 seems to simply recover the gradient $\nabla_x F^t(x) - F^l(x)$ and thus, Alg.2 standard gradient descent. Can you comment on this?
2) Why did you choose $x_i = 3.3 \times 10^9$ for Figure 2b? From reading just the text, one would assume $x_i \in [-1, 1]$, making this slightly misleading. This also seems to be the source for the upper bound on floating-point errors cited in the intro. Given that this work mostly claims to be the first to show attacks on unperturbed networks and regions around test-set images, this seems misleading.
3) Your conclusion that “with the increasing rounding error, a greater leeway is left between the real certified radius R and the computed certified radius $\tilde{R}$ for our method to exploit, so the attack success rate increases.” seems to contradict the results of Figure 2a), which show that while the rounding error in computing $||\delta||$ becomes increasingly more pronounced as dimensionality is increased (Gap between strong and weak threat model), the attackability under the strong threat model (ignoring this error) steadily decreases.
4) What precision was used to compute the robust radii for linear SVMs? And similarly what precision is used to compute the perturbation norm in Section 4.3? Can you clarify what settings are used for Gurobi, in particular, what precision and feasibility tolerances?
5) For positive inputs, any ReLU network with positive weights and biases is linear, under these circumstances, linear bound propagation methods such as $\beta-CROWN$ are exact (up to floating point errors) and should thus not be considered “conservative”. Similarly, for ReLU networks, verification via MILP is complete, thus while the obtained certificate is not “complete” using the definition in this work, it should not be considered “conservative”. Does your approach also work for settings where the analysis is actually conservative? E.g. using $\beta$-CROWN on a network with 2 hidden layers of 20 Neurons each?
7) Could you discuss why you expect verification (not norm computation) with interval arithmetic to reduce the success rate of your attack to 0% under the weak threat model? It seems that the norm computation should still be attackable in the weak setting (unless interval arithmetic is also used there, which should be clarified). Possibly, analysis with interval arithmetic merely introduces enough conservativeness to prevent this type of attack.

### Comments
* When using a citation marker as a grammatical element of a sentence, I suggest using "\citet" and otherwise "\citep" to comply with the author guide section 4.1.
* I would suggest you rewrite the second bullet point in core contributions to more clearly distinguish the two threat models.
* Typically, a different definition of completeness of robustness verifiers is used in the literature. That is, a method is called complete if it can prove every robustness property that holds (see e.g. your closest related work Jia and Rinard (2021)). I would consider adapting this definition or at least highlighting this difference.
* Python permits exact rational arithmetic and infinite precision floating point real arithmetic with suitable packages. I would consider this when discussing the strong threat model in Section 3.
* Should $R$ in the first paragraph of Section 4 not be $\tilde{R}$ ?
* I suggest using different line types in Figure 2 a to make W and S easier to distinguish
* Table captions should be placed above a table not below to comply with the author guide Section 4.4.

### Conclusion
In summary, while the goal of this paper to demonstrate the vulnerability of neural network verification methods to floating-point attacks on unmodified networks around test-set samples to be interesting and novel, I believe this claim is not sufficiently substantiated by the conducted experiments. Only a single non-linear neural network is considered and only attacked successfully under the "weak threat model" which merely demonstrates that the used norm computation can be attacked. Further, there are substantial questions regarding the novelty of core parts of the paper, with Alg. 1 and 2 seeming to recover standard PGD and the suggestion of using interval arithmetic for verification already being established 5 years ago.

---

> ### Author Response · Authors · 2023-11-22
>
> $\mathbf{Weakness}$
>
> $\mathbf{Ans.1}$
> ReluPGD is only half of our attack (in the case of ReLU nets),
> and it can be replaced by any standard approach to finding directions for adversarial perturbations.
> As the name was already suggesting, ReluPGD is indeed an instantiation of PGD.
> However we acknowledge that we can better highlight this (obvious) connection.
> Our inclusion of ReluPGD (Algorithms 1 and 2), derived by hand,
> was to emphasize that it is exact for a local linearization.
> The choice of ReLU network was two-fold: (1) it is an important architecture in practice
> (2) it permits strong attacks. ReluPGD is intended to transparently highlight this second point.
> Hand derivation is by no means an interesting contribution, which is why the details are relegated to the appendix.
> The second part of our attack (Algorithm 3) is where we innovate
> by leveraging the structure of certificates to isolate floating-point rounding problems.
> This simple configuration of the step size is enough to demonstrate violations of certification guarantees.
>
> $\mathbf{Ans.2}$
> Thank you for listing these works. We will add them to the paper.
> Singh et al. (2018 and 2019) also use interval arithmetic in neural network certification, and are sound w.r.t. floating-point arithmetic.
> However, Singh et al. (2018 and 2019) use the $\ell_{\inf}$ metric in neural network certification,
> which avoids the influence of potential rounding errors in the calculation of the perturbation norm.
> We use the $\ell_2$ metric, and handle the impact of rounding errors in the calculation of
> the perturbation norm for certification in the strong threat model.
>
> Voráček and Hein (2022) attacked randomized smoothing by also exploiting the finite representation of floating points.
> However, they attacked synthetic data and model, while we show attacks on unperturbed networks and test-set images.
>
> $\mathbf{Ans.3}$
> For random linear classifiers in Section 4.1,
> we use both 32-bit and 64-bit floating-point precision, as shown in Fig 2(a).
> For Linear SVM, we use 64-bit floating-point precision.
> For neural nets, we use 32-bit floating-point precision.
> Gurobi tolerance is $2\times10^{-5}$.
>
> $\mathbf{Ans.4}$
> In the strong threat model, we used the upper bound of the perturbation norm $\overline{\|\boldsymbol{\delta}\|}$, not the real norm $\Delta$.
> The difference between the weak ($\tilde{R}-\|\boldsymbol{\delta}\|$) and
> strong ($\tilde{R}-\overline{\|\boldsymbol{\delta}\|}$) models is not the floating-point soundness of the norm computation.
> It can be the floating-point soundness of the norm computation if we attack ($\tilde{R}-\Delta$) in the strong threat model instead.
>
> $\mathbf{Ans.5}$
> For MILP, we verify two 3-layer neural network multiclass classifiers with 100 nodes in their hidden layer,
> one of them is trained normally, whose parameters (weights and biases) can be negtive.
> For Randomized Smoothing, we also attack a 3-layer network with negtive parameters.
> The verification from MILP could potentially be conservative,
> and the verification from Randomized Smoothing should mostly be conservative.
>
> $\mathbf{Questions}$
>
> $\mathbf{Ans.1}$
> Please refer to the answer of W.1.
>
> $\mathbf{Ans.2}$
> We want to show how big rounding errors could potentially be in Figure 2b.
> In the paper, we do state that the rounding errors exploited by our attack are small.
> Please see the second paragraph of Section 5:
> “For example, the rounding error for the certified radius of the first MNIST image of Figure 3 (Appendix A) is in the 13th decimal place”.
>
> $\mathbf{Ans.3}$
> In the strong threat model, we use the upper bound of the perturbation norm $\overline{\|\boldsymbol{\delta}\|}$ estimated with interval arithmetic.
> With increasing dimension, more operations are involved, interval arithmetic gets more uncertain (loose) about its estimation,
> and outputs a bigger interval that contains the real result.
> That is, the upper bound $\overline{\|\boldsymbol{\delta}\|}$ gets bigger (estimated more loosely).
>
> In all, with increasing dimension, $\tilde{R}$ gets bigger, but $\overline{\|\boldsymbol{\delta}\|}$ also gets bigger.
> Hence, the leeway $\tilde{R}-\overline{\|\boldsymbol{\delta}\|}$ exploited by our attack in strong models
> first gets bigger ($D\le18$) and then gets smaller ($19\le D\le100$),
> and our success rate first increases then decreases.
>
> $\mathbf{Ans.4}$
> Please refer to the answer of W.3.
>
> $\mathbf{Ans.5}$
> Please refer to the answer of W.5.
>
> $\mathbf{Ans.6}$
> Our mitigation introduces theoretically enough conservativeness (i.e., $\gamma = \tilde{R} - \underline{R}$)
> by providing $\underline{R}$ as a robustness certificate.
> $\underline{R}$ is safe in the strong threat model.
> In the weak threat model, this $\gamma$ will not have same provable guarantees. We will clarify this.

---

> > ### Comment · Reviewer_GFuC · 2023-11-23
> > **Thank you for the response**
> >
> > I greatly appreciate the authors' efforts in addressing my questions, however several of my key concerns are not sufficiently addressed:
> >
> > Weaknesses: A1, A2, A4, A5
> > Questions: A2, A3
> >
> > I thus maintain my score.

---

### Official Review · Reviewer_musS · 2023-10-31

**Soundness:** 3 good
**Presentation:** 3 good
**Contribution:** 3 good
**Rating:** 8
**Confidence:** 4

**Summary:**

This manuscript pokes a hole on robustness certificates due to the implementation issue. That  is, the floating point representations could result in unsound certifiable radius for many methods as listed in this paper. In general this is a system bug in the certification process.

**Strengths:**

This paper is well-motivated and well-written. The problem it identifies is novel and also critical to the application of certifiable robustness in real-world systems. Float32 is probably the standard one we will use, and in the real-world we will have more quantizations for resource constraints. If this bug needs to be gone only with float128 or even more, I don’t think we would actually deploy the certification method.

**Weaknesses:**

There are no significant weaknesses in this paper. One minor thing is that it only experiments with linear models and pretty small networks. The impact of this bug could be much more serious if the authors can experiment with larger datasets and larger models. Furthermore, there is no discussion on non-ReLU models and Lipschitz-based certification methods. Does the bug also appear in those models? Can the attack generalize the MinMax models? I did not see real issues why one could’t try to find unsound points for non-ReLU networks. Can the authors explain the blockers here? With that being said, I would appreciate the authors to have a paragraph to discuss what the limit of the attack they find. The family of methods will be affected and the family of methods that won’t be affected. This would add great value to the current paper.

**Questions:**

My questions are included in the Weakness part already.

---

> ### Author Response · Authors · 2023-11-22
>
> $\mathbf{Ans.1}$
> Verification methods such as $\beta$-CROWN and MIP tend to
> give tighter estimates of robustness certificates for smaller and simple neural nets.
> That is, those methods can verify a relatively bigger (tighter) $\tilde{R}$ for a small model,
> which results in a bigger leeway $\tilde{R}-R$ for our attack to exploit and allows higher attack success rates.
> For larger neural nets, the certificates could be more conservative,
> but larger neural nets (more operations) could also entail larger rounding errors.
> We thank the reviewer for this suggestion. We hope that any violations of the intended certificate by a mechanism on any sized network would demonstrate the issue to researchers in the area. We hope that before future deployments researchers communicate potential implementation flaws to end users, or adopt mitigations like those proposed here.
>
>
> $\mathbf{Ans.2}$
> Though in theory the attacks should still hold, attacking MinMax models and Lipschitz-based certificates in practice
> is an interesting direction for future work.
> We chose ReLU nets owing to it being (1) used frequently in practice
> (2) being readily linearizable, with the intention of simply demonstrating a range of certificate violations.
>
> Our current limitations were summarised in the submission in Appendix G (being in the supplemental it may have been missed). We identify the following limitations, that could also be addressed as future work.
> First, our attacks against randomized smoothing do not study the relationship
> between the attack success rate and the soundness probability of these methods due to their probabilistic nature.
> A possible future direction would be to study how the two interact.
> Second, we showed how to adopt our mitigation based on interval arithmetic generally,
> and demonstrate this mitigation for linear models and exact certifications.
> Detailed exploration of embedding these mitigations in other verifiers and models is left as future work.

---

> > ### Comment · Reviewer_musS · 2023-11-22
> > **Thanks**
> >
> > Thanks for there reply. These responses should go into the main body of the paper. Similarly, the limitation of the work should be placed in the main body as well (there is no reason to put them into the Appendix).

---

### Official Review · Reviewer_s866 · 2023-11-01

**Soundness:** 2 fair
**Presentation:** 2 fair
**Contribution:** 1 poor
**Rating:** 1
**Confidence:** 5

**Summary:**

Certified defenses against adversarial attacks for ML classifiers provide a mathematical upper bound $R(x)$ on the amount of corruption needed to misclassify a given input $x$. This paper studies the discrepancy between the radius $\bar R(x)$ output by an implementation of such defenses on systems supporting finite-precision arithmetic, say $1.0$, and the _real_ radius $R(x)$ guaranteed by the theory, say $0.999999$, demonstrating adversarial perturbations that _break_ such certified defenses by exploiting floating point approximations.

**Strengths:**

The paper highlights a potentially overlooked problem with trusting certified defenses too much: the finite precision implementation of such defenses on computers might lead to slightly optimistic certificates. Hence, this paper highlights the fact that practitioners critically relying on such systems must use a marginally lower value of the certified radius than reported.

**Weaknesses:**

1.	Relevance: The greatest weakness of this paper is the lack of relevance of the problem studied. Specifically, it is very well known that computers perform finite-precision arithmetic, and for any computation (much beyond adversarial robustness), the results might have a minuscule margin of error due to the floating point approximations involved. In the light of this, any application that requires exact precision must implement explicit safeguards (e.g. interval arithmetic) to protect against rounding problems. For the case of adversarial examples, it would be good if the authors could provide examples of applications where an application would care about getting the certified radius exactly right (this is not true for instance on many commonly studied applications like image classification, object detection, etc.).

2.	Trivial Solution?: A naïve fix to the above problem, in the context of certified robustness, which the paper also mentions on P2, is that for super-sensitive applications, one can simply report $R(x) - \gamma$ as the certificate for a small $\gamma$, instead of $R(x)$ to ensure that the output certificate is valid. $\gamma$ can be really small, for instance, P8 mentions that for MNIST, the approximation errors are at the 13th decimal point. The paper argues that finding a fixed $\gamma$ is not possible in general, but fails to provide convincing evidence on any real tasks. On a synthetically constructed task, $\gamma$ is showed to be potentially large (0.1), but then the scale of $R$ is orders of magnitude larger leading to the same solution. Generally, depending on the task at hand, there should be a principled way to choose $\gamma$ depending on scale of the error that can be tolerated for the problem.

3.	Proposed method exponential in dimension?: The proposed approach to finding adversarial perturbations, is roughly to perturb each dimension of the input a bit till a misclassification is achieved. A slightly smarter way of doing these floating point flips is utilized, but the approach is essentially exponential in dimension. How much time does the proposed algorithm take per image? from P8, it seems like a time-out of 15 minutes is set for every image.

4.	Proposed Method ReluPGD same as PGD applied to max-margin objective? Could the authors comment on how the ReLUPGD algorithm differs from a standard PGD attack using the max-margin loss (e.g., loss(positive class) - loss(negative class) for the binary case).
5.	Proposed Method’s applicability in real scenarios: In practice, most certification methods for Neural Networks produce certificates that are very pessimistic for most input points. In light of this, it is hard to believe that a very high decimal place attack would even be successful for real networks. The examples shown in this work linearize the network or study synthetic problems with linear boundaries, and this effect does not show up.

6.	Writing: The writing could be improved at several places, and some portions could be cut to make way for the core contributions:
	1.	P2 Contribution 1: “invalidate the implementations” —  invalidate might be a too strong word here.
	2.	P2 Contribution 2: “floating point norm” is not defined till now.
	3.	Statement unclear: “in the case where the library computes …”, which library are we talking about here?
	4.	P2 Contribution 3: Cohen et. al.’s main certificate is probabilistic.
	5.	P3: No reference needed for distance of a point to a plane.
	6.	P4: The main contribution, the “neighbors” algorithm could be moved to the main text for clarity, currently it is unclear what this is.
	7.	P5: It is unclear whether the entire warmup section is needed, since the end result is a PGD-like algorithm, and PGD is quite standard in the literature now.
	8.	P5: The algorithm ReLUPGD should appear in the main paper, since it is one of main contributions.
	9.	Fig 2, P6: For the synthetic experiments, what is the setup, is $x, w, b$ fixed and $D$ is changing? If so, how were these values decided? Are there any average statistics over these values?
	10.	P9: Theorem 1 is a standard statement of interval arithmetic, and as such it is unclear whether it should be a theorem.

**Questions:**

Questions are mentioned above alongside the weaknesses.

---

> ### Author Response · Authors · 2023-11-22
>
> $\mathbf{Ans.1}$
> The reviewer is right that it is well-known that flaws are known in regards to finite-precision arithmetic.
> However, this observation was neither made before against exact certification mechanisms nor shown to work against conservative mechanisms.
> Our work has demonstrated robustness violations against a wide range of certifications on unperturbed networks and test-set images,
> and reminds practitioners that if certified robustness is to be used for security-critical applications,
> their guarantees and implementations need to account for limitations of modern computing architecture.
>
>
> $\mathbf{Ans.2}$
> We introduced the idea of a radius correction $\tilde{R}-\gamma$ as a straw argument:
> it is meant to be flawed. We hope the reviewers and AC understand that non-zero attack success rate is not acceptable.
> This is because such a mechanism (with constant $\gamma$) would not be sound and introduce uncontrolled errors.
> We showed in Section 5 that we can still find violations if we set certificate as $\tilde{R}-0.1$.
>
> The reviewer asks if there is a principled way to find $\gamma$. The only approach known to us is input dependent
> (non-constant) $\gamma = \tilde{R} - \underline{R}$, such that $\tilde{R}-\gamma = \underline{R}$,
> where $\underline{R}$ is the lower bound of the certified radius estimated with interval arithmetic,
> and we propose to use it in our mitigation as the certificate.
>
> $\mathbf{Ans.3}$
> We avoid exponential runtime by adopting random sampling in the neighboring search algorithm as follows.
> We randomly sample $N$ floating-point neighbors of the adversarial instance $\mathbf{x'}$ to
> explore robustness violations close to the decision boundary due to rounding errors.
> For example, $N$ is 5000 in Section 4.2.
>
>
> $\mathbf{Ans.4}$
> ReluPGD is only half of our attack (in the case of ReLU nets),
> and it can be replaced by any standard approach to find directions for adversarial perturbations.
> As the name was already suggesting, ReluPGD is indeed an instantiation of PGD.
> Our inclusion of ReluPGD (Algorithms 1 and 2), derived by hand,
> was to emphasize that it is exact for a local linearization.
> The choice of ReLU network was two-fold: (1) it is an important architecture in practice
> (2) it permits strong attacks. ReluPGD is intended to transparently highlight the second point.
> Hand derivation is by no means an interesting contribution, which is why the details are relegated to the appendix.
>
> There maybe confusion from Reviewer s866 on where our contribution lies
> (i.e., it is not a new general-purpose PGD algorithm).
> For example the second part of our attack (Algorithm 3) is where we innovate
> by leveraging the structure of certificates to isolate floating-point rounding problems.
> This simple configuration of the step size is enough to demonstrate violations of certification guarantees.
>
>
> $\mathbf{Ans.5}$
> Verification methods such as $\beta$-CROWN and MIP tend to give tighter estimates of robustness certificates for smaller neural nets.
> That is, those methods can verify a relatively bigger (tighter) $\tilde{R}$ for a small model,
> which results in a bigger leeway $\tilde{R}-R$ for our attack to exploit and allows higher attack success rates.
> We experiment with small neural nets that are normally trained and have 100 neurons in the hidden layer on the real MNIST dataset in Section 4.3,
> and show that even if the robustness certificates are potentially conservative,
> we can still find violations for those that are relatively tight, although not exact.
> For larger neural nets, the certificates could be more conservative,
> but larger neural nets (more operations) could also entail larger rounding errors, so a successful attack is still possible.
> Moreover, our attack exploits subtle floating-point errors, as we explained in the second paragraph of Section 5:
> For example, the rounding error for the certified radius of the first MNIST image of Figure 3 (Appendix A) is in the 13th decimal place.
>
> The purpose of our work is to show that certifications can be attacked.
> The fact that all the certification mechanisms we examined are unsound should be of serious concern.
> There is nothing to indicate that higher levels of rounding (and hence rounding attacks)
> wouldn't be possible in other datasets, or domains. Without sound mechanisms,
> one cannot trust implementations to perform as advertised.
>
>
> $\mathbf{Ans.6}$
> For the library, we refer to the library that computes the square root for the norm
> and represents it as an interval, i.e., PyInterval in our study.
> For Fig 2, we experiment with models of different dimensions, with $D$ ranging from 1 to 100,
> as we want to explore how the dimension influences our attack.
> One dot in Fig 2 stands for an experiment on a model of a certain dimension (e.g., $D=50$).

---

> > ### Comment · Reviewer_s866 · 2023-11-22
> >
> > Thanks for the responses. As other reviewers pointed out, the relevance of this study is unclear in light of simple fixes (non-constant $\gamma$ as simply a small constant fraction of $R$), and existence of prior work as well as general intuition around the floating point problem (the authors say that prior work study other norms or synthetic settings, however the  experiments in the present work are arguably pretty toyish), and unclear experimental choices (scale of $R$ vs $\gamma$, lack of aggregate statistics). In light of these issues, I will keep my score.

---

### Official Review · Reviewer_FP18 · 2023-11-01

**Soundness:** 3 good
**Presentation:** 2 fair
**Contribution:** 2 fair
**Rating:** 5
**Confidence:** 3

**Summary:**

This paper proposes a new attack methodology to break robustness certificates given by existing methods. It exploits the rounding errors present in the floating point computation of the certificates. The authors test the efficacy of their attack on linear classifiers, linear SVM, and neural network models and empirically show that such exploits exist for all these models. Finally, the authors propose a formal mitigation approach based on bounded interval arithmetic to prevent such exploits.

**Strengths:**

- The high success rate of the proposed attack shows the magnitude of the problem and the need for it to be addressed.
- The initial proposed solution might help spark some more research for making existing certification more robust to such exploits.

**Weaknesses:**

- While the paper states that it is not easy to fix the problem by just replacing the certified radius $\tilde{R}$ with $\tilde{R} - \gamma$, ($\gamma << 1$), it is not clear what the attack success rate looks like when a $\gamma$ that is a small fraction of $\tilde{R}$ is used. As the strong threat model already leads to a huge drop in attack success, it seems plausible that just certifying a slightly more conservative radius might suffice for practical use (as the bounds in some cases like Randomized Smoothing are already probabilistic).
- The proposed method is not shown for most of the state-of-the-art certification methods. It would be great to see the effective drop in certified radius that happens when using the proposed method and see the computation cost as well.

**Questions:**

Please refer to the Weaknesses section for the questions.

---

> ### Author Response · Authors · 2023-11-22
>
> $\mathbf{Ans.1}$
> We introduced the idea of a radius correction $\tilde{R}-\gamma$ as a straw argument: it is meant to be flawed.
> While we understand that Reviewer FP18 is interested in attack success rates for small constant $\gamma$,
> we hope the reviewers and AC understand that non-zero attack success rate is not acceptable.
> This is because such a mechanism (with constant $\gamma$) would not be sound.
> We showed in Section 5 that we can still find violations if we set certificate as $\tilde{R}-0.1$;
> with increasing dimension there is increased opportunity for undesirable rounding.
>
> Reviewer FP18 highlights the probabilistic nature of certificates for Randomized Smoothing:
> while it is true that such mechanisms have a failure probability,
> this probability is still controlled and only accounts for sampling error in the Monte Carlo estimates
> made by Randomized Smoothing. Any non-zero failure rate due to rounding (which is inevitable with constant $\gamma$)
> represents a greater failure rate, that is no longer controlled in any way.
> Moreover, for the vast literature on deterministic certificates,
> the goal is a consistent prohibition of adversarial examples.
> Our proposed mitigation (based on interval arithmetic) uses $\underline{R}$ as a robustness certificate.
> This can be understood as a principled way to choose an input-dependent (non-constant) $\gamma = \tilde{R} - \underline{R}$,
> such that $\tilde{R}-\gamma = \underline{R}$.
> Hence, interval arithmetic provides a sound approach with a provable attack success rate of zero.
>
> $\mathbf{Ans.2}$
> We report experiments on representative certification mechanisms of the Exact, Conservative and Randomized types.
> For the latter two types we used some of the seminal works.
> For example, MIP certification used for neural networks in Section 4.3 is known for its tight
> estimate for robustness certificates, although its slow running speed on big models is a problem.
> Nevertheless, our MIP certifications on small neural networks
> give tight estimates for robustness certificates with acceptable runtime, and allow our attack to have high success rates.
> Though it is interesting to see our methods against all certification methods,
> our contribution is against the fundamental observation that theoretical (real arithmetic based) radius does not match the one computed in practice.